# Dimensioning of Cycle Lanes Based on the Assessment of Comfort for Cyclists

**Darja Šemrov *** **, Robert Rijavec and Peter Lipar**

Faculty of Civil and Geodetic Engineering, University of Ljubljana, Jamova Cesta 2, 1000 Ljubljana, Slovenia
* Correspondence: darja.semrov@fgg.uni-lj.si; Tel.: +386-41-605-104

**Abstract:** In a century where mobility is becoming more sustainable in terms of energy transition, emissions reduction, and a healthy quality of life, the use of bicycles is increasing and has many advantages over other modes of transport that have been underused. The bicycle is an excellent alternative for short distances of up to five kilometers. In combination with public transportation, it can also successfully compete with motorized transport for longer distances. For the adequate development of cycling, it is necessary to create the right conditions in terms of accessibility and road safety. This means planning appropriate cycling infrastructure where cyclists feel comfortable and safe, which can lead to additional increased use in bicycles for everyday trips. Comfort for cyclists is a concept supported by road safety, a pleasant environment, connectivity, and the attractiveness of cycling infrastructure. In other words, cyclists respond to the physical, psychological, and sociological aspects of the cycling experience that are also related to the cycling infrastructure and environment: where I am, what I see and perceive, and how I feel. This paper presents the concept of the level of service for cyclists (BLOS) as a unified method for defining the comfort of cyclists. This paper presents the method for determining the level of service or comfort for bicyclists as a function of road width, width of the cycling area, traffic volume, and the speed and structure of motorized traffic flow. The result of BLOS, the mathematical model used, is graphically presented and allows decision-makers and designers of cycling infrastructure to easily assess the suitability of cycling infrastructure. Different diagrams for different input data are presented in the paper.

**Keywords:** road safety of cyclists; cycling lane; level of service for cyclists

## 1. Introduction

Cycling is widely recognized as one of the most environmentally friendly modes of transport, having a positive impact on the health of individuals and, indirectly, on the health of society as a whole. For this reason, many people around the world support the development of cycling infrastructure and promote cycling as a mode of transport (especially in urban areas). Cycling is also becoming more popular in Slovenia, and policymakers are following this trend by improving conditions for cycling and providing cycling infrastructure. A best-practice example is the expansion of cycling infrastructure in Ljubljana, which was included in the Copenhagenize Index—the most comprehensive and holistic ranking of bicycle-friendly cities—in 2015 and has been on this list ever since. Unfortunately, as the number of cyclists grows, Slovenia faces a trend of increasing accidents involving cyclists, up by 2% in 2018 and 8% in 2019. In 2020, the number of accidents decreased by 1%, and one less cyclist died than in 2019, but it is important to note that population mobility decreased in 2020 due to COVID-19 pandemic measures. In the last five years, the percentage of seriously injured cyclists increased by 15% [1]. In the last five years, 24% of serious injuries in road accidents were cyclists (the highest percentage of all modes), which is a clear signal that we need to pay special attention to cyclist safety when planning cycling infrastructure.

Despite potential danger and accidents, numerous studies have been conducted to demonstrate the benefits of cycling. Researchers find that cycling is associated with a lower risk of health problems [2,3] and that regular cycling improves mental and physical health [4,5]. In addition to the positive effects on individuals, the effects on the environment have also been shown to improve, as cycling instead of using a car or public transport reduces emissions of $CO_2$, CO, $NO_x$, and PM particles and reduces noise pollution. In addition, cyclists (and pedestrians) have positive impacts on the local economy and tourism [6].

Due to the positive impacts of cycling noted above, planners and policymakers are working in a variety of ways to improve the quality of the cycling experience, thereby increasing the number of trips made by cycling. Numerous research has shown that cycling infrastructure plays a key role in the choice of cycling as a mode of transport, so it is important to provide (potential) users with a safe cycling infrastructure.

As can be seen from the literature review (please refer to Section 2), researchers use different input data and different methods when answering the question of why and where cyclists ride bicycles. Due to the different research approaches and different cultures in different cities, the results of the studies cannot be compared, and the proposed measures cannot be implemented in different countries without critical evaluation. Therefore, the aim of this paper is to standardize the assessment of cycling infrastructure and make the results comparable between different countries.

In the field of motorized traffic, the level of service (LOS) with the limited upper acceptable (qualitative) value at the end of the planning period (in the case of Slovenia, the planning period is 20 years after the renewal/upgrading/construction of the road or intersection) is an integral part of the traffic study and a decisive criterion among various solutions considered. Considering the need to harmonize urbanization and road infrastructure development for motorized traffic, bicyclists, and pedestrians and to ensure the sustainable development of cities and rural areas, the paper proposes not only an assessment of the LOS for motorized traffic, but also an assessment of the level of service for bicyclists (BLOS) at the end of the planning period. A consistent methodology for evaluating BLOS will ensure adequate cycling infrastructure not only at the time of implementation, but also at a time when motorized traffic characteristics might change. In other words, one can plan the right size of cycling infrastructure for future traffic or take other administrative measures to maintain motorized traffic volumes and/or heavy vehicle mode share at a level that ensures comfortable cycling.

This paper presents the results of a parametric study of BLOS [6] considering measurable input data (characteristics of motorized traffic), and the graphical representation of the results can be used as an additional requirement for planning and designing cycling infrastructure.

The paper is organized as follows. Section 2 reviews previous studies and approaches to assessing the reasons for cycling and the impact of cycling infrastructure on cycling mode share. Section 3 provides a formal description of the harmonized and unified assessment of cycling infrastructure using the BLOS methodology. Section 4 reports on the proposed BLOS assessment taking into account Slovenian legislation, which provides room for improvement in the planning and design of cycling infrastructure and provides a graphical representation of the BLOS parametric study. Finally, Section 5 summarizes the contribution of the paper and suggests future research.

## 2. Literature Review

The review of research and approaches to assessing the reasons for cycling and the impact of cycling infrastructure on cycling mode share will identify factors and decision elements that need to be considered to harmonize and standardize the assessment of cycling infrastructure using the BLOS methodology discussed in the next section.

In the literature, the influence of cycling infrastructure on mode choice, i.e., the decision by the user to use a bicycle or another means of travel and on route choice has

been studied from different perspectives. Researchers have demonstrated a relationship between choosing a bicycle instead of a vehicle for shorter trips, examining travel time [7] and distance [8,9]. Researchers have also studied the effects of motorized traffic, showing that users (do not) choose to cycle based on the traffic load and speed [10] and that the characteristics of motorized traffic influence cycling route choice [11,12]. Research shows that cyclists are willing to ride longer distances if they believe it is safer [13,14] and that cyclists perceive lower traffic load and speeds as safer and less stressful [15,16].

Researchers study the effects of choice of route on the decision to cycle and recommend the most appropriate type of cycling infrastructure based on the research results. The research can be divided into two groups. The first group of research includes those that monitor the habits of cyclists using GPS devices [8,9,17–21] or through preferences expressed in questionnaires and interviews [17,18] or through a combination of both approaches [12]. The second group of research focuses on quantifying different factors, e.g., different types of cycling infrastructure [22] and the influence of the lateral distance that motor vehicle drivers use to overtake cyclists riding on cycle lanes [10,23–26], on the perceptions of cyclists on safety.

All of the above studies aim to determine which cycling infrastructure is most commonly used and which cycling infrastructure encourages cycling. The results of the studies are understandably inconsistent and depend on the degree of the cycling culture in each country and the input data and methodology used, etc. The common denominator in all studies is that the decision to cycle and the choice of route is influenced by many factors: the available cycling infrastructure and motorized traffic; the volume and share of heavy vehicles; the built environment; and the personal perception of safety, stress, comfort and attractiveness of the surroundings. It is evident that there is a wide range of proposed methods, complex data collection, subjective responses, and a large number of variables that do not facilitate or simplify the work of decision-makers in defining and implementing cycling infrastructure. For this reason, a standardized, validated, and unambiguous assessment method, the proposed BLOS, was called for to create a scenario that is directly derived from (and influenced by) the responsibilities of decision-makers (as described in Section 4).

On the other hand, it can be stated that cyclists need to feel safe in the traffic system, and there is a general conclusion that cycling infrastructure separated from motorized traffic attracts a greater number of cyclists. Research has shown that it is important for cyclists to have cycling infrastructure that is separated from motorized traffic by at least a dividing line, but it is even more appreciated when the cycling infrastructure is completely separated from other traffic, including pedestrians [8,9,17,20,27–31]. Since the construction of a cycling infrastructure separated from motorized traffic in urban areas is difficult and at the same time requires a large financial investment, a cycle lane is often constructed. As mentioned above, it is important to provide adequate cycling infrastructure for cyclists, especially when there is a higher potential for conflict with motorized traffic. This paper proposes a method for determining the width of cycle lanes based on motorized traffic characteristics by calculating BLOS.

## 3. Level of Service for Cyclists

In this chapter, the description of the BLOS starts from the central focus that the width of the cycle lane determines the level of service for cyclists. The width of a cycling infrastructure depends primarily on the average daily traffic (ADT), the direction distribution of traffic, the type of cycling infrastructure, and the speed and structure of motorized traffic, as described below.

The concept of BLOS could be defined as harmony between cyclists and the environment due to the balance of physical, psychological, and sociological aspects. Assessment is complex because it is based on the characteristics of each person and is therefore very subjective. In the literature, researchers refer to different terms that describe the harmony between cyclists and the environment: pleasant cycling, the suitability of cycling, and cycling-friendliness of the environment. In addition, different research areas address differ-

ent aspects of cycling comfort, namely, road safety, pleasant environment, connectivity, and attractiveness of cycling areas, all of which contribute to cycling comfort [32]. The diversity of terminology can lead to misunderstandings.

Although the assessment of the BLOS is a mathematical function of human stimulus perception, it can be similarly described in terms of measurable physical attributes of motorized traffic and road conditions. The entire process was conducted with a high degree of statistical confidence. The equation for the level of service for cyclists provides an estimate of the discomfort and ambiguity:

1. maximum traffic flow next to the cycling infrastructure;
2. road width, speed limit, and percentage of freight traffic;
3. condition of the cycling infrastructure surface;
4. width of the cycling infrastructure.

The statistically calibrated equation used to calculate the level of service for cyclists is adjusted to evaluate cycling conditions in a common, uniform road environment. The same measurable traffic and road factors are used as in other traffic modules. With statistical accuracy, the model reflects the impact on the suitability or comfort cyclists will experience due to factors such as road width, cycling infrastructure width, traffic volume, cycling infrastructure conditions, permitted speed, the percentage of heavy vehicles (trucks and buses), and longitudinal parking.

The equation for calculating the level of service for the BLOS of cyclists is [33,34]:

$$BLOS = 0.507 \cdot ln\left(\frac{Vol_{15}}{n_{RL}}\right) + 0.199 \cdot K_V(1 + 10.38 \cdot HV)^2 + 7.066 \cdot \left(\frac{1}{P}\right)^2 - 0.005(w_{RL}^* + w_{CL})^2 + 0.760 \quad (1)$$

where:

$Vol_{15}$ 15 min traffic volume $Vol_{15} = \frac{ADT \cdot D \cdot K_D}{4 \cdot PHF}$:

$ADT$ average daily traffic

$D$ directional factor

$K_D$ peak to daily factor

$PHF$ peak hour factor

$n_{RL}$ number of lines for motorized traffic

$K_V$ effect of speed

$K_V = 1.1199 \cdot ln(V - 20) + 0.8103$

$V$ permitted speed for motorized traffic

$HV$ percentage of heavy vehicles (trucks and buses)

$P$ five-point pavement surface condition rating of FHWA

$w_{CL}$ average width of cycle lane

$w_{RL}$ average width of lane for motorized traffic

$w_{RL}^*$ average effective width of lane for motorized traffic

$w_{RL}^* = w_{RL}$, if $ADT > 4000 \ veh/day$

$w_{RL}^* = w_{RL}(2 - 0.00025 \cdot ADT)$, if $ADT \leq 4000 \ veh/day$

The resulting estimate of the level of traffic comfort for cyclists resulting from this equation is divided into service categories "A, B, C, D, E, and F" according to the ranges shown in the table below (Table 1). This reflects the perceptions of the bicycle users on the level of service on the road section. Grades between A and D represent appropriate traffic comfort.

**Table 1.** Coherence between numerical estimates and service categories.

| Service Categories | BLOS |
|---|---|
| A | $\leq 1.5$ |
| B | $>1.5$ and $\leq 2.5$ |
| C | $>2.5$ and $\leq 3.5$ |
| D | $>3.5$ and $\leq 4.5$ |
| E | $>4.5$ and $\leq 5.5$ |
| F | $>5.5$ |

## 4. Planning and Designing of Cycling Infrastructure: The Slovenia Case Study

In this section, in order to describe the factor(s) that influence the BLOS and the areas of competence (or intervention) of the decision-makers to whom the BLOS is directed, we will define the interaction between the types of cycling infrastructure and their interaction with the BLOS using the criteria for selecting the type of cycling infrastructure.

### 4.1. Types of Cycling Infrastructure

Comparable to other countries, Slovenian rules on cycling areas define several types of cycling infrastructure, namely, cycle paths, cycle tracks, cycle lanes, cycle lanes on sidewalks, cyclists not separated from pedestrians on sidewalks, the advised safety lane and sharrow. In the following, the definitions of each type of cycling infrastructure are summarized, with the widths of the cycling infrastructure and the restrictive conditions of motorized traffic, as stated in Slovenian legislation.

The cycle path (Figure 1) is a stand-alone infrastructure at least 2.5 m wide that is primarily intended for cycling but in certain cases may also be intended for other users, such as pedestrians, motor vehicles, and agricultural machinery.

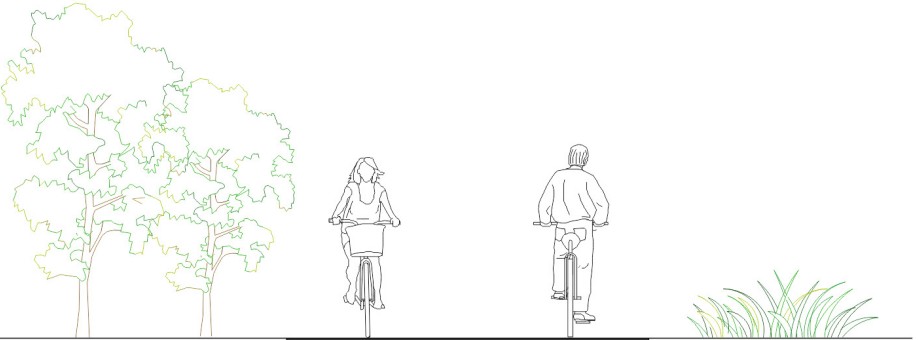

**Figure 1.** Cycle path.

The cycle track (Figure 2) is a cycling infrastructure separated from the carriageway for bicycles and mopeds with a speed of no more than 25 km/h and a width of no less than 1.0 m. The cycle track must be separated from the carriageway by a curb height of at least 10 cm and a safety width of 0.5 m at the speed limit V $\leq$ 50 km/h, 0.75 m at V $\leq$ 70 km/h or 1.0 m at speeds up to 90 km/h. If the cycle track is at the same level as the carriageway, a safety width of at least 1.0 m must be ensured.

If the spatial conditions do not enable the construction of separate cycling infrastructure, a cycle lane at least 1.0 m wide (recommended 1.75 m) (Figure 3) can be designed. A cycle lane is part of the carriageway and is only separated from motorized traffic by a solid white and red line.

In addition to the basic types of cycling infrastructure presented above, Slovenian legislation also permits cycle lanes on sidewalks, cyclists not separated from pedestrians on sidewalks, advised safety lanes, and sharrow.

The advantages and disadvantages of each type of cycling infrastructure are summarized in the following table (Table 2).

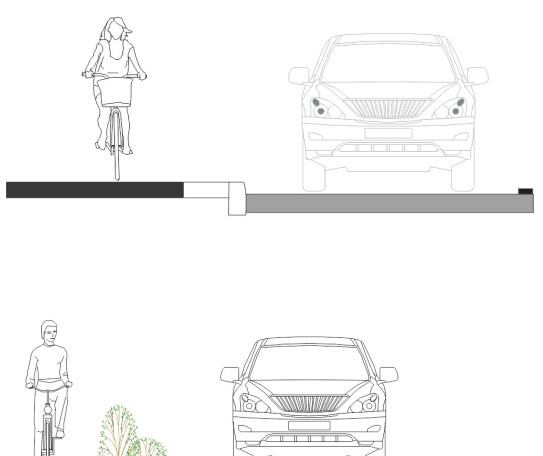

**Figure 2.** Cycle track.

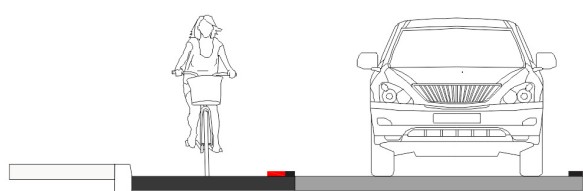

**Figure 3.** Cycle lane.

**Table 2.** Advantages and disadvantages of different types of cycling infrastructure.

| Type of Cycling Infrastructure | Advantages | Disadvantages |
|---|---|---|
| cycle path | away from traffic<br>greater connectivity<br>high safety and comfort | require a lot of space<br>expensive |
| cycle track | physical separation from traffic<br>high level of traffic safety<br>easier overtaking of other cyclists | a high level of risk due to accesses to building and right-turn movement of vehicles, and higher speeds of road users and thus less attention to cyclists<br>higher demand for space<br>financially demanding |
| cycle lane | financially more sustainable solution<br>requires less space than the construction of a cycle path or track<br>increased visibility of cyclists at intersections and thus greater safety (proper solutions are required)<br>easy and fast implementation | no physical separation from vehicles<br>gives drivers the feeling that they do not need to keep special attention on cyclists |
| advised safety lane | easy and fast implementation<br>financially favorable solution | increased risk for cyclists<br>cyclists have less chance of overtaking |
| sharrow | no need for additional space<br>more spatially and financially favorable solution | increased risk for cyclists<br>cyclists obstruct motor traffic<br>cyclists have less chance of overtaking |
| cycle lanes on sidewalks | requires less space | increased potential for conflicts between pedestrians and cyclists<br>greater risk for vulnerable road users, e.g., children and elderly |

The focus of the paper is on the cycle lanes, since cyclists ride alongside motorized traffic and are therefore at higher risk of accidents, and, at the same time, cycle lanes are commonly used as a cost-effective solution. According to national regulations, the cycle lane should be 1.0 m wide in Croatia and 1.5 m in Germany. It should be at least

1.25 m (recommended 1.5 m) for permitted speeds below 50 km/h and at least 1.5 m (recommended 1.75 m) for speeds of motorized traffic above 50 km/h. It is worth noting that in the country with the highest percentage of bicycle use, the Netherlands, the absolute minimum width of bicycle lanes was increased to 2.0 m in the spring of 2022, and the recommended width is now 2.3 m. The paper aims to raise awareness of the importance of an appropriate width of cycling lane using the BLOS method.

### 4.2. Criteria for Selecting the Type of Cycling Infrastructure

According to Slovenian national legislation rules on cycling areas, the type of cycling infrastructure is selected based on the maximum speed limit and hourly traffic volume of motorized traffic along with the cycling infrastructure, as shown in the following figure (Figure 4). In Zone I, bicyclists are allowed to ride on the carriageway together with motor traffic (advised safety lane and sharrow); in Zone II, cyclists must be provided with at least a cycling lane; and in Zones III and IV, cyclists must be provided with infrastructure separated from motorized traffic (cycle paths and cycle tracks).

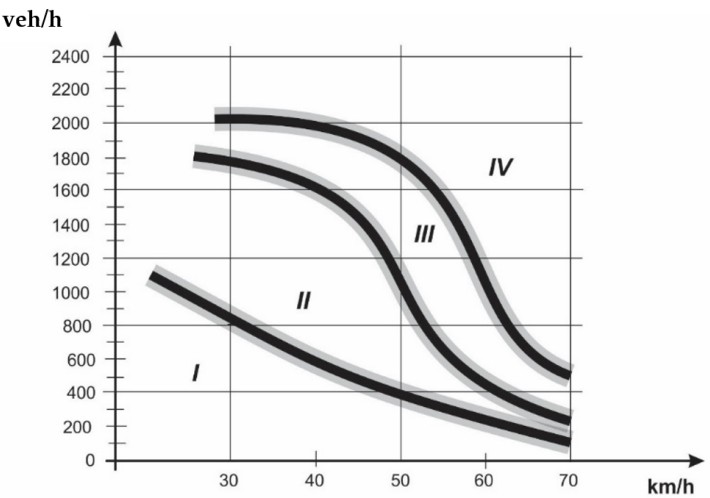

**Figure 4.** Criteria for cycling infrastructure selection.

Figure 4 shows that, according to Slovenian national legislation, on a road with a traffic load of 1100 vehicles per hour and a speed limit of 50 km/h, the cycle lane or cycle track can be chosen. If the latter is chosen, a safety width of 0.5 m between the roadway and the bike lane must be provided on the sidewalk, which already physically separates motorized traffic and cyclists and must be at least 1.0 m wide. In the case of a cycle lane, cyclists are separated from motorized traffic only by a line, and the width provided for them can also be as little as 1.0 m.

In national legislation, the width of a cycle lane is given in the range from 1.0 m to 1.75 m, where the criteria for choosing the width of the cycle lane is not specified. Since motorized traffic, especially heavy vehicles such as trucks and buses, running along the cycle lane has a significant impact on the safety of cyclists due to wind gusts [13,35], and it should also not be ignored that according to data from the USA as much as 2% of all accidents are due to rollover crashes caused by bad tires [36], this study focuses on the impact of traffic volume and traffic structure on the width of the cycle lane through the assessment of BLOS.

### 4.3. Methodology

The research focused on analyzing traffic conditions, traffic volumes, traffic structure, and cyclist comfort in the open section, so the analysis does not include intersection areas.

As indicated in HCM, a road near the center of an urban area often has a D-factor near 50% with traffic volumes equal for both directions and recommends defaults of PHF-

factor 0.92 for urban facilities, and the peak to daily factor is set at 0.1; these values for the D-factor, PHF-factor and KD-factor were used in the mathematical model, BLOS. Since the maintenance of cycling infrastructure should be regularly done to ensure the safety of cyclists, the research assumed the cycle line pavement surface was in good condition ($p = 4$).

For different combinations of traffic volumes (ADT between 100 and 20,000, step 100) and percentages of heavy vehicles (between 0 and 20%, steep 0.1%) the BLOS according to Equation (1) was calculated. Based on the numerical results and categories presented in Table 1 the boundaries between the BLOS categories are presented.

*4.4. Results*

The first analysis is, in a way, a generalization of the problem, as the analyses take into account the most common speed limit in urban areas of 50 km/h and the corresponding road width of 2.75 m. Since the recommended cycle lane width in Slovenia is set at 1.75 m, the results of the analyses refer to this width.

The diagram in Figure 5 allows decision-makers and designers of cycling infrastructure to graphically evaluate the level of service for cyclists as a function of motorized traffic characteristics, i.e., traffic volumes (ADT) and share of heavy vehicles, where we recommend cycling infrastructure with elements that provide a level of service of at least E at the end of the planning period. Traffic volumes and the percentage of heavy vehicles considered in the evaluation of cycling infrastructure should be carefully estimated using state-of-the-art methods for traffic forecasting.

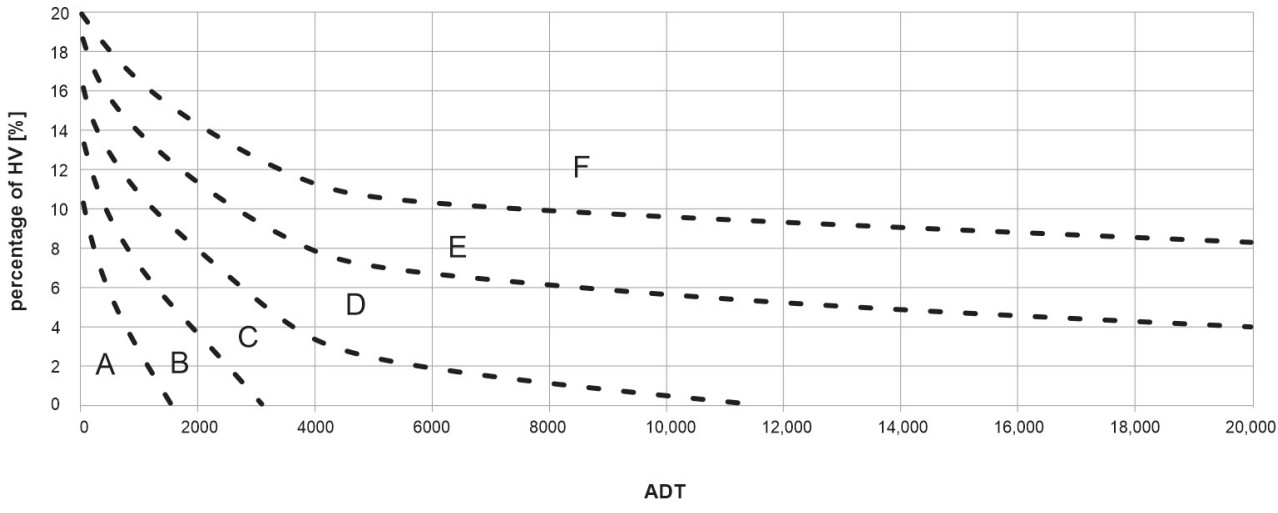

**Figure 5.** Level of service for $V = 50\ \frac{km}{h}$, $w_{RL} = 2.75$ m, $w_{CL} = 1.75$ m, where A–F are service categories.

Since the width of cycle lanes is not strictly defined in Slovenian legislation (and, at the same time, the transferability of the results is ensured, since different countries have different rules for the width of cycling lane), the second part of the research focuses on the selection of appropriate widths of cycle lanes depending on the traffic volume and the share of heavy vehicles, separately, for the speeds V = 50 km/h and V = 70 km/h, in order to achieve a level of service for cyclists of at least E (BLOS < 5.5) and at least D (BLOS < 4.5). The calculation took into account the most commonly used lane width for motorized traffic in Slovenia, namely 2.75 m for roads with a maximum speed of 50 km/h and 3.00 m for roads with a speed of 70 km/h.

From the results, it can be concluded that to ensure a comparable level of traffic comfort for cyclists, a wider bike lane should be provided in case of less favorable parameters of motorized traffic (higher ADT and/or higher percentage of heavy vehicles) or in case of higher speeds of motorized traffic.

The results show that not only traffic volume but also the characteristics of motorized traffic (percentage of heavy vehicles) must be taken into account because an increase in the percentage of trucks by only 2% means doubling the width of the cycle lane to ensure a comparable level of service for cyclists.

Slovenian national legislation specifies the minimum width of the cycling lane at 1.0 m. Based on the results of the analyses, it can be concluded that cycle lanes 1.0 m wide should not be installed on roads with a speed limit of V = 50 km/h and a truck share of more than 8–10%, except for ADT lower than 4000 vehicles. On roads with a speed limit of 70 km/h with ADT higher than 4000 vehicles, the percentage of trucks and buses should not exceed 6–8%. If the cyclist should be given a better sense of safety, in other words, a higher level of traffic comfort (BLOS of at least D), the percentage of trucks should be reduced to 3–6% at speeds up to 50 km/h and 2–4% at speeds up to 70 km/h.

## 5. Conclusions and Future Work

This work has demonstrated the relationship between the criteria used to select different types and characteristics of cycling infrastructure and the level of service for cyclists. This relationship is validated by BLOS, which can be used in a harmonized and validated way to support decision-makers in the planning, design and management of these infrastructures.

Indeed, researchers use different methods to identify the characteristics of cycling infrastructure that have a positive impact on the number of cyclists, either by tracking cyclists using GPS devices or through questionnaires. The research results differ in the details, but, in general, we can conclude that the number of cyclists is influenced by the sense of safety of the cyclists.

In the research, the concept of the level of service for cyclists was analyzed, which allows the assessment of the comfort of cycling and its classification into six categories (A to F). The assessment of the level of traffic comfort for cyclists allows designers to determine the appropriate width of cycling infrastructure. In the research, we focused on the cycle lane, which is part of the carriageway and only separated from it by a dividing line. The graphically presented results of the analysis (Figures 6–9) can be used as a tool and supplement to the currently applicable national regulations in determining the width of the cycle lane, taking into account the total traffic volume and the percentage of heavy vehicles.

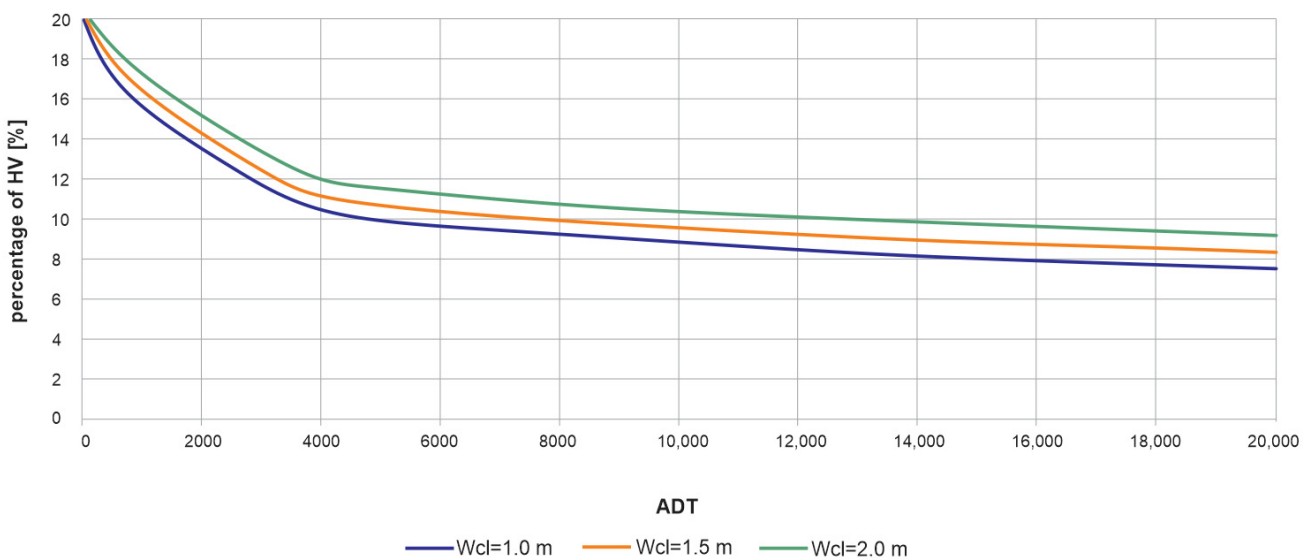

**Figure 6.** Width of cycling lane to provide BLOS of at least E if $V = 50$ km/h, $w_{RL} = 2.75$ m.

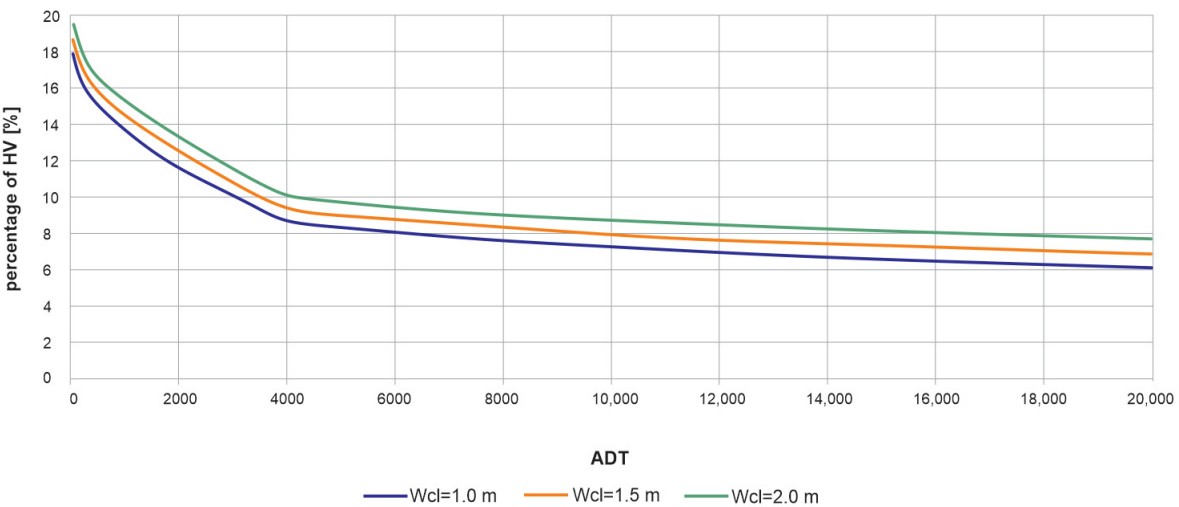

**Figure 7.** Width of cycling lane to provide BLOS of at least E if $V = 70$ km/h, $w_{RL} = 3.00$ m.

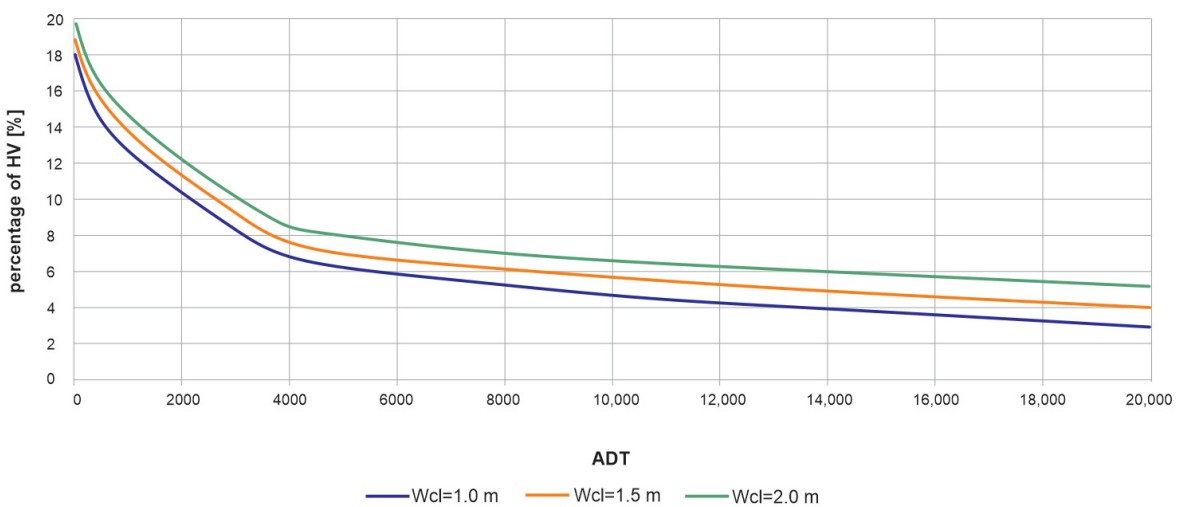

**Figure 8.** Width of cycling lane to provide BLOS of at least D if $V = 50$ km/h, $w_{RL} = 2.75$ m.

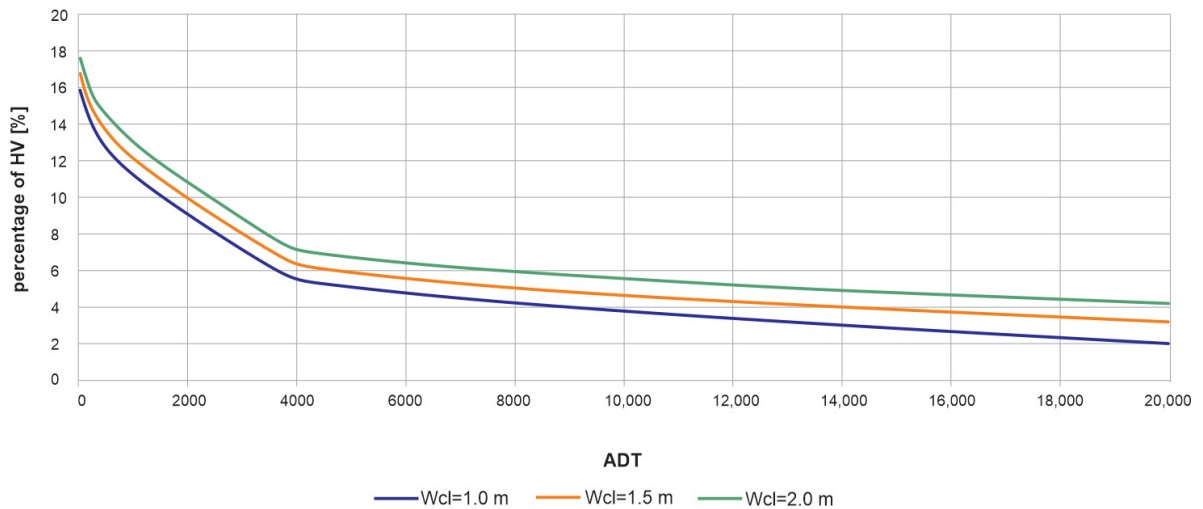

**Figure 9.** Width of cycling lane to provide BLOS of at least D if $V = 70$ km/h, $w_{RL} = 3.00$ m.

In the present research, using the mathematical model suggested, coefficients were used. Therefore, in the continuation of the research, we will focus on calibrating the equation to the Slovenian driving culture, and in later stages, we will investigate the method of determining the traffic comfort for cyclists at intersections.

**Author Contributions:** Methodology, D.Š. and P.L.; writing—original draft preparation, D.Š. and R.R.; writing—review & editing, P.L. All authors have read and agreed to the published version of the manuscript.

**Funding:** This research received no external funding.

**Conflicts of Interest:** The authors declare no conflict of interest.

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
