# Peer review of "Dimensioning of Cycle Lanes Based on the Assessment of Comfort for Cyclists"

_sustainability, doi:10.3390/su141610172_

Round 1

Reviewer 1 Report

I find this a highly unclear paper. The aim is not clear. The scientific contribution is not made clear. The paper contains no methods section in which they explain what methods they have applied to meet their goal (which is not clear). The conclusion is very vague in my view.

In my view, this paper needs to be rewritten in a far clearer way. With a clear goal, clear methods, and a mature discussion of what the scientific contribution is: what do they add to the scientific knowledge base in this field? (so relate their findings to the literature, section 2).

More detail:

Line 12: this could double the use of bicycles? Really? How determined?

The goal of this paper is unclear. I think that ‘This paper presents the process of determining the level of service or comfort 17 for cyclists depending on the width of the road, the width of the cycling area, the volume of traffic, 18 the condition of the road and the speed and type of motor vehicles. Why is it relevant to present this process? For whom relevant? What is the scientific relevance of doing this?

In the intro, a goal or research question is lacking.  

I do not understand how their literature review (section 2) is connected to this paper. What is the knowledge gap they have identified? And which aim to fill with this paper.

Section3 suddenly starts with ‘Slovenian rules’. Why? If this paper I about the Slovenian context, I think this should be introduced (and explained why) already in the introduction

As it is not clear what they aim for in this paper I find it difficult to understand what I am reading in sections 3 and 4.  I also miss a methods section in this paper. I understand that they want to present the process. What methods did they apply to present and analyse this process? Interviews?

I find the results unclear. I do not see a process presented at all. I see different figures for the width of cycle lanes. Why is this relevant for the scientific community? Is this new for the scientific community? What is their contribution exactly?

The conclusion is very vague (and contains all kinds of new information). 

Author Response

Response to the reviewer

Manuscript sustainability-1768685 entitled Dimensioning of cycle lane based on assessment of the (in)comfort for cyclist

Let us first note that we appreciate the careful consideration of our manuscript, detailed comments and useful suggestions provided by the reviewers. The authors would like to thank the editor and the reviewers for considering our manuscript interesting. Please find specific point-by-point responses to the individual comments and requests for revisions below.

Line 12: this could double the use of bicycles? Really? How determined?

Although some researchers reported a doubling of share of cyclists in cities, we agree with the comment that the statement is too general; in the revised manuscript, we refer to "additional increased use of bicycles for everyday trips."

The goal of this paper is unclear. I think that ‘This paper presents the process of determining the level of service or comfort for cyclists depending on the width of the road, the width of the cycling area, the volume of traffic,  the condition of the road and the speed and type of motor vehicles. Why is it relevant to present this process? For whom relevant? What is the scientific relevance of doing this?

The authors added an additional explanation about the research objective and the added value of the graphical presentation of the results - a tool for decision-makers and cycling infrastructure designers to determine the appropriate width of a cycling lane. As explained in the literature review, the researchers did a good job formalizing perceptions of bicycling infrastructure and answering why and where people bike. To harmonize the descriptions, this study proposes the mathematical model BLOS, which could serve as a common denominator in selecting the width of the cycle lane. In the Netherlands, a 2.3 m wide cycle lane is standard, but in developing countries, more or less 1 m is reserved for cyclists, which can lead to accidents or to cycling infrastructure not being used as it could be.

In the intro, a goal or research question is lacking.  

In Section “Introduction” goal and relevance are stated more clearly.

I do not understand how their literature review (section 2) is connected to this paper. What is the knowledge gap they have identified? And which aim to fill with this paper.

The literature review section aims to describe different approaches and methods that lead to the same results - different factors influence cyclists, but the studies cannot be compared to each other or are not transferable to other countries without expert knowledge, and the studies do not give precise answers on the recommended type and/or width of cycling infrastructure. This can be achieved with the proposed mathematical model.

Section3 suddenly starts with ‘Slovenian rules’. Why? If this paper I about the Slovenian context, I think this should be introduced (and explained why) already in the introduction

We have reconsidered the comment, and paper is carefully rewritten and emphases Slovenian aspect with the title of section 4.

As it is not clear what they aim for in this paper I find it difficult to understand what I am reading in sections 3 and 4

We thank you for the comment - we have carefully rewritten the paper, also taking into account the reviewers' comments to make the paper clearer and easier to understand.

I also miss a methods section in this paper. I understand that they want to present the process. What methods did they apply to present and analyse this process? Interviews?

Description added to section 4.3 Methodology.

I find the results unclear. I do not see a process presented at all. I see different figures for the width of cycle lanes. Why is this relevant for the scientific community? Is this new for the scientific community? What is their contribution exactly?

As mentioned above, the scientific community has done great work and proposed several methodologies, but the main idea of the paper is to a) harmonize the assessment of cycling infrastructure, b) highlight the importance of the width of cycling infrastructure (in this case, cycling lanes), as this is a problem (at least) in Slovenia. 

The conclusion is very vague (and contains all kinds of new information). 

The conclusion section is rewritten.

Reviewer 2 Report

The paper analyzes the influence of traffic volume and traffic structure on the bicycle lanes' width and evaluates the level of service and traffic comfort of cyclists.

The paper is interesting; however, I have several concerns:

Figure 1 is redundant and the width of 2.5 is incorrect in many places in the world.

The safety distances in the beginning of page no. 3 are baseless and unsupported by evidence.  

The paper discusses the Slovenian national legislation, but are these sizes more or less than the standard sizes in other places in the world?

The authors do not take into account rollover accidents which should be tackled by a larger safety distance. In Y. Wiseman, (2010, July), "Take a picture of your tire!", Proc. of 2010 IEEE International Conference on Vehicular Electronics and Safety (pp. 151-156). Available online at: https://u.cs.biu.ac.il/~wisemay/icves2010.pdf  The author explains about rollover accidents caused by poor tires. I would encourage the authors to cite this paper and explain how the potential rollover accidents influence the decision on safety distances.

Author Response

Response to the reviewer

Manuscript sustainability-1768685 entitled Dimensioning of cycle lane based on assessment of the (in)comfort for cyclist

Let us first note that we appreciate the careful consideration of our manuscript, detailed comments and useful suggestions provided by the reviewers. Authors would like to thank the editor and the reviewers for considering our manuscript interesting. Please find specific point-by-point responses to the individual comments and requests for revisions below.

Figure 1 is redundant and the width of 2.5 is incorrect in many places in the world.

The paper has been rewritten to explain in more detail that the descriptions and widths of cycling infrastructure refer to the Slovenian definitions in the document "Slovenian rules on cycling areas" - 2.5 m is the Slovenian standard for the width of cycle path. We agree with the comment that this is "wrong" width.

The safety distances in the beginning of page no. 3 are baseless and unsupported by evidence.

We could not agree more with this remark, hence the research presented in this paper. Unfortunately, these distances are specified in the Slovenian regulations.

The paper discusses the Slovenian national legislation, but are these sizes more or less than the standard sizes in other places in the world?

The paper focuses on the cycle lane, and the widths are not standardized, please see last paragraph of section 4.1. Types of cycling infrastructure

The authors do not take into account rollover accidents which should be tackled by a larger safety distance. In Y. Wiseman, (2010, July), "Take a picture of your tire!", Proc. of 2010 IEEE International Conference on Vehicular Electronics and Safety (pp. 151-156). Available online at: https://u.cs.biu.ac.il/~wisemay/icves2010.pdf  The author explains about rollover accidents caused by poor tires. I would encourage the authors to cite this paper and explain how the potential rollover accidents influence the decision on safety distances.

Thank you for the suggestion, the proposed manuscript does indeed explain why a wider safety distance (please note that in the revised manuscript we refer to the safety width) is an important factor to be considered.

Round 2

Reviewer 2 Report

The authors made a decent effort.